# Air Pollution Affecting Pollen Concentrations through Radiative Feedback in the Atmosphere

**Carsten A. Skjøth** [1,*], **Alexander Kurganskiy** [1,2], **Maria Grundström** [1,3], **Małgorzata Werner** [4] **and Beverley Adams-Groom** [1]

1    School of Science and the Environment, University of Worcester, Henwick Grove, Worcester WR2 6AJ, UK;
     A.Kurganskiy@exeter.ac.uk (A.K.); dr.m.grundstrom@gmail.com (M.G.);
     b.adams-groom@worc.ac.uk (B.A.-G.)
2    College of Life and Environmental Sciences, University of Exeter, Treliever Road, Penryn,
     Cornwall TR10 9FE, UK
3    Swedish Meteorological and Hydrological Institute, Folkborgsvägen 17, 60176 Norrköping, Sweden
4    Department of Climatology and Atmosphere Protection, University of Wrocław, ul. Kosiby 8,
     51-621 Wrocław, Poland; malgorzata.werner@uwr.edu.pl
*    Correspondence: c.skjoth@worc.ac.uk; Tel.: +44-(0)1905-85-5226

**Abstract:** Episodes with high air pollution and large amounts of aeroallergens expose sensitive individuals to a health damaging cocktail of atmospheric particles. Particulate matter (PM) affects the radiative balance and atmospheric dynamics, hence affecting concentrations of pollutants. The aim of the study is to estimate feedback between meteorology and particles on concentrations of aeroallergens using an extended version of the atmospheric model WRF-Chem. The extension, originally designed for PM and dust, concerns common aeroallergens. We study a birch pollen episode coinciding with an air pollution event containing Saharan dust (late March to early April 2014), using the model results, pollen records from Southern UK and vertical profiles of meteorological observations. During the episode, increased concentrations of birch pollen were calculated over the European continent, causing plumes transported towards the UK. The arrival of these plumes matched well with observations. The lowest parts of the atmospheric boundary layer demonstrate a vertical profile that favours long distance transport, while the pollen record shows pollen types that typically flower at another time. The model calculations show that feedback between meteorology and particles changes pollen concentrations by ±30% and in some cases up to 100%. The atmospheric conditions favoured meteorological feedback mechanisms that changed long distance transport of air pollution and aeroallergens.

**Keywords:** pollen; air pollution; meteorology; feedback-effects; aeroallergens; modelling

## 1. Introduction

Allergic rhinitis (AR), caused by bioaerosols with allergenic contents, is estimated to negatively affect 400 million people worldwide and more than 300 million from asthma [1]. Symptoms from AR negatively impact the quality of life [2,3] and patients with AR often suffer from asthma or develop asthma later in life [4], formulated through the hypothesis of one airway one disease [5]. Aeroallergens are just a fraction of bioaerosols [6] and aeroallergens that typically have much lower atmospheric concentrations than traditional air pollutants such as $NO_x$ and particulate matter (PM) [7]. The economic burden on society from aeroallergens is substantial and in Europe it was estimated that just one aeroallergen, ragweed pollen, costs Euro 7.4 billion annually [8].

Air pollutants are known to affect the potency of aeroallergens [9,10]. Many studies indicate a need to consider both pollen species and standard air pollution for the epidemiological evaluation of environmental determinants in respiratory allergies [10–12]. Large geographical variations in allergenic potency were demonstrated [13,14]. Furthermore,

pollen from urban areas and more polluted regions have higher amounts of allergens even though the amount of pollen is the same [15]. High levels of chemical air pollution (e.g., $SO_2$) and PM, e.g., from desert dust, have been observed along with long distance transport (LDT) of aeroallergens [16]. LDT is generally episodic [17] and has been observed for many different aeroallergens for a number of decades [18–20]. Similar, episodes with desert dust, e.g., from Sahara, are also irregular. During such episodes, the atmospheric conditions favour LDT of particles that are emitted into the air masses whether this is particulate matter (e.g., $PM_{10}$) or bioaerosols (e.g., pollen). Tree pollen such as oak (*Quercus*) and olive (*Olea*), dominate in southern Spain [21] in early April. In the UK, early April often marks the start of the birch (*Betula*) pollen season [22], while oak usually peaks several weeks later [23]. This suggests that during an air pollution episode with transport of desert dust, the airstream may pick up pollen from different species during northwards transport. Furthermore, desert dust episodes have been shown to cause a radiative feedback on mesoscale meteorology [24] that intensify aerosol pollution [25]. A change in mesoscale meteorology impacts air mass transport and hence LDT of particles. This suggests that air pollution episodes, in particular those with high particulate matter associated with transport of desert dust, could simultaneously coincide with LDT of several different aeroallergens and impact mesoscale meteorology, significantly affecting the concentrations of the aeroallergens involved. We test this hypothesis through the use of an atmospheric model handling both air pollutants, bioaerosols and feedback, based on an extension of WRF-Chem [26,27]. We do this by calculating the impact of feedback during a desert dust episode 29 March–1 April 2014 and focusing on the United Kingdom. We identify the existence of potential LDT transport of several bioaerosols by exploring all available observations of allergenic pollen in central and southern England. Finally, we combine this with an analysis of the atmosphere focusing on its vertical structure during the episode using both observations and model calculations with and without feedback.

Commonly, the aerosol radiative feedback and properties are studied using observational in-situ and satellite data [28–30] as well as atmospheric dispersion models [31,32]. The usage of atmospheric dispersion models allows a separation of local and LTD components of air pollutants but has rarely been conducted on bioaerosols (pollen). Using WRF-Chem also provides an estimate of the aerosol radiative feedback effect on meteorology and hence the LTD component of pollen concentrations. This, in turn, highlights the advantage and necessity of our approach with respect to hypothesis testing. Finally, to the author's knowledge the study is the first ever attempt quantifying the effect of aerosol feedback on pollen concentrations using atmospheric dispersion models.

## 2. Materials and Methods

### 2.1. Sites and Observation Methods of Pollen and Vertical Atmospheric Structure

The pollen observations in this study are obtained from five sites (Figure 1): Plymouth (50.3544, −4.1199), Worcester (52.19670, −2.2421), the Isle of Wight (50.7111 −1.3009), Cambridge (52.2116, 0.1349) and Ipswich (52.0561, 1.1984). Data from Plymouth, the Isle of Wight, Cambridge and Ipswich are obtained from the background monitoring programme operated by the UK Met Office. Meteorological observations are from the two sounding sites, Herstmonceux (50.90″ N, 0.32″ E) and Camborne (50.22″ N, 5.32″ W), also shown in Figure 1.

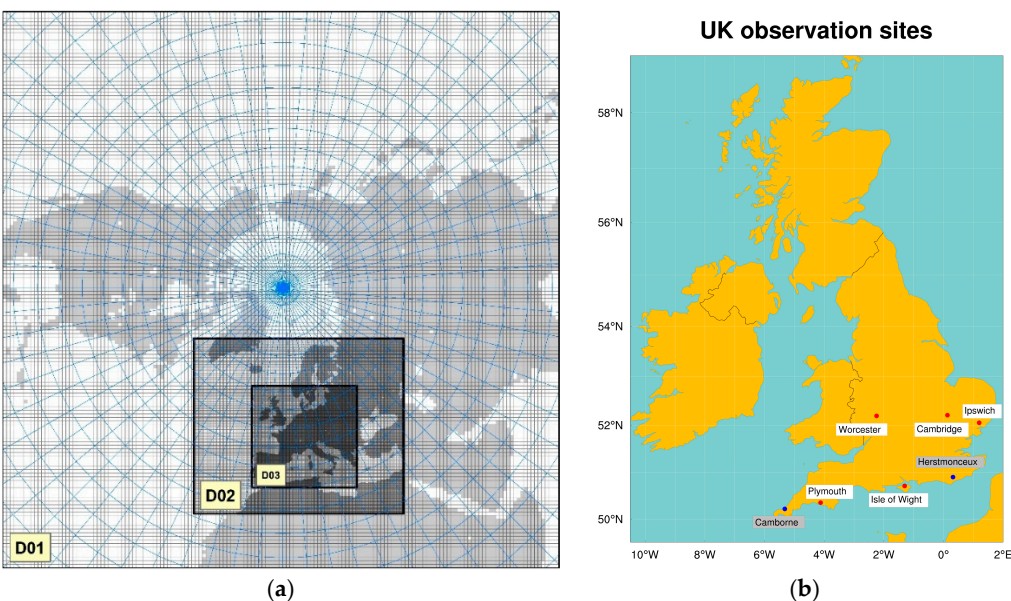

**Figure 1.** Left (**a**) showing the geographical setup of the three model domains in the WRF-Chem extension with 108 km (D01), 36 km (D02) and 12 km (D03) resolution (model results provided from the inner domain 3). Right (**b**) showing the UK with the location of the 6 pollen monitoring sites (red dot and white background text) and the 2 sites with radio-soundings (blue dots, grey background text) used in the study.

Both pollen monitoring sites and atmospheric sounding stations are located in southern England (Figure 1) which is an area with limited elevation, typically 0–300 m [22]. Therefore, this area has only limited impact on mesoscale meteorology caused by hills and mountains. The area has substantial amounts of small woodlands with important pollen-producing trees including Alder (*Alnus*), Birch (*Betula*) and Oak (*Quercus*) [22,33] To the South and East the area is bordered by the North Sea and English Channel, with the European continent on the other side. Pollen was recorded at all six sites using a Burkard volumetric pollen and spore trap of the Hirst design [34], a method that has been used for decades in the UK monitoring network [23]. Pollen is identified at the genus level according to Käpyla and Penttinen [35], here covering the eight trees Hazel (*Corylus*), Alder (*Alnus*), Willow (*Salix*), Birch (*Betula*), Ash (*Fraxinus*), Elm (*Ulmus*), Oak (*Quercus*) and Plane (*Platanus*). Pollen measurements are provided as daily mean concentrations (pollen/m$^3$) according to international standard recommendations [36]. We analyse the entire pollen record during the period 26 March–4 April 2014 (Table 1). Observations of the vertical structure of the atmosphere are obtained through atmospheric soundings, which are provided by the on-line platform maintained by University of Wyoming (Laramie, WY, USA) [37]. Here we extracted height above the surface and vertical wind speed profiles for the central part of the episode: 29 March–1 April 2014. We used all available wind speed observations from the surface and up to 3500 m, which typically corresponds to 22–30 records. Observations are available at the main synoptic observational time, i.e., 00 UTC.

**Table 1.** Observed pollen concentrations in 2014 at the Isle of Wight (IoW), Worcester (Wor), Plymouth (Ply), Cambridge (Cam) and Ipswich (Ips) for *Betula* (birch) and *Quercus* (oak).

| Date | *Betula* **Sp.** | | | | | *Quercus* **Sp.** | | | | |
|------|-----|-----|-----|-----|-----|-----|-----|-----|-----|-----|
| **25/3** | **IoW** | **Wor** | **Ply** | **Cam** | **Ips** | **IoW** | **Wor** | **Ply** | **Cam** | **Ips** |
| 26/3 | 1 | 0 | 0 | 5 | 27 | 0 | 0 | 0 | 0 | 0 |
| 27/3 | 3 | 1 | 0 | 1 | 63 | 0 | 0 | 0 | 0 | 0 |
| 28/3 | 1 | 4 | 0 | 6 | 319 | 0 | 0 | 0 | 0 | 0 |
| 29/3 | 21 | 24 | 1 | 27 | 326 | 0 | 0 | 0 | 0 | 0 |
| 30/3 | 113 | 310 | 21 | 330 | 76 | 0 | 1 | 1 | 0 | 0 |
| 31/3 | 27 | 330 | 24 | 149 | 432 | 0 | 0 | 1 | 0 | 0 |
| 1/4 | 8 | 147 | 1 | 142 | 1562 | 0 | 0 | 1 | 0 | 0 |
| 2/4 | 191 | 188 | 72 | 114 | 702 | 0 | 0 | 1 | 0 | 1 |
| 3/4 | 329 | 528 | 49 | 765 | 223 | 0 | 0 | 6 | 0 | 0 |
| 4/4 | 356 | 49 | 7 | 534 | 194 | 0 | 0 | 0 | 0 | 0 |
| 5/4 | 28 | 303 | 23 | 231 | 27 | 0 | 0 | 0 | 0 | 0 |

*2.2. Geographical Setup of the Model Calculations and Choice of Parametrisations in WRF-Chem*

The applied model is WRF-Chem [38,39] version 3.5 which was modified with a number of extensions (see next section). Here we applied the model using a polar stereographic projection with 3 domains (Figure 1) of 108, 36 and 12 km resolution, respectively. This geographical setup ensures that large-scale features leading to the transport of Saharan dust are captured whilst the focus area (UK) has sufficient high geographical resolution, i.e., 12 km. Previous studies in relation to pollen, atmospheric dynamics and emissions processes identified 36 km as insufficient, while 12 km seems to work well in many cases [40,41]. The WRF-Chem model simulations start on the 23rd of March to ensure sufficient spin-up (3 days) before the Saharan dust episode. The model is applied with two different scenarios: with and without radiative feedback. The WRF part of the model has a similar setup as Skjoth et al. [40] with 48 vertical layers and an increased number of layers near the surface. Inputs to the model are daily sea surface temperatures and FNL (Final) global analysis data maintained by the National Centre for Environmental Predictions (NCEP), having a spatial resolution of $1° \times 1°$ (longitude–latitude), a temporal coverage of 6 h, and a vertical resolution of 27 pressure levels. Each new layer is nudged into WRF-Chem. The most important parameterisations are the following physical options: The rapid radiative transfer model (RRTM) for longwave radiation [42], the Dudhia scheme for shortwave radiation [43] the Kain Fritsch Scheme for cumulus parameterization [44] the Yonsei University scheme for PBL physics [45] and the NOAH land-surface model [46]. This choice of parameterisations follows the same setup of parameterisations as Skjøth et al. [40] which also corresponds to well-tested studies using the WRF or WRF-Chem model [47,48] with the exception of the Yonsei University scheme, which replaces the MYNN scheme [49] as the MYNN scheme according to the WRF-Chem manual has not yet been extensively tested with the GOCART aerosol module.

*2.3. Extensions to the WRF-Chem Model*

The GOCART (Georgia Tech/Goddard Global Ozone Chemistry Aerosol Radiation and Transport) module available in WRF-Chem v. 3.5 has been extended by adding 10 extra variables, complementing the existing bins in WRF-Chem. Each variable corresponds to one of 9 different pollen types grouped at the genus level such as *Betula* sp., *Quercus* sp., *Ambrosia* sp. and *Alnus* sp., while the 10th variable is a duplicate of one of the other variables with respect to physical properties, in this case *Betula* sp. This allows for simultaneous runs of emission scenarios that can separate pollen sources (see next section) and run the model along with both $PM_{2.5}$, $PM_{10}$ emissions as well as dust from natural sources, in this case focusing on how the atmosphere transports pollen (and dust) during a known Saharan dust episode. Dry and wet depositions of pollen particles in the atmosphere are included in the WRF-Chem model by taking into account standard sizes

of pollen grains with respect to the aerodynamic properties of spherical particles. In this case we are focusing on birch, which has been given a particle diameter of 20 μm and a density of 1000 kg/m$^3$. Dry deposition in WRF-Chem contains both resistance analogy and gravitational settling [50] here extended with the physical properties of the pollen. Wet deposition uses the scavenging coefficient approach by Jung and Shao [51], which is both relatively simple and provides similar results to other commonly applied schemes [51]. The wet deposition scheme takes into account collection efficiency of aerosol particles up to 40 μm by different raindrop sizes up to 3000 μm. The original model code for wet deposition was obtained from the WRF-Chem v. 3.9 and was numerically optimised for efficient runs in WRF-Chem and then extended with the 10 additional variables covering different pollen types.

### 2.4. Emission to the WRF-Chem Model: PM and Pollen

Commonly, WRF-Chem emissions are based on external emission models and, in the case of anthropogenic emissions, they are typically based on fixed emission factors. In order to increase flexibility of the extensions and their geographical applicability, we have taken a different approach by incorporating a pollen emission module that is directly connected to the WRF Preprocessing System (WPS) and the hourly meteorological variables in WRF/WRF-Chem. We added static maps to the WPS system such as tree cover, separated into broadleaved and conifer trees and tree species maps that relate to either conifer or broadleaved tree maps. The tree cover has been re-gridded to a global data set with ~0.017° resolution (a 40,320 × 16,354 lat-lon grid) obtained from a combination of the Globcover data set [52] and CLC—Corine Land Cover [53]. We are here using the CLC data set whenever possible as it provides the best representation of the two with respect to smaller woodlands that are found in the UK and other European countries [22]. The seasonal phenological model for birch flowering is the one described by Skjøth et al. [40], originally developed for the Danish Eulerian Hemispheric Model [54] where it is calibrated to the pollen season for Worcester 2014. Hourly pollen release is activated by sunlight when it exceeds 1000 W/m$^2$ and deactivated during rain (0.5 mm/h) using an exponential decrease that removes 20% of the available pollen mass/h during optimal conditions. This pattern corresponds well to the typical increase in birch pollen concentrations as seen in Northern Europe such as Worcester, London and Copenhagen, Denmark, where hourly birch pollen concentrations typically increase rapidly after 9 AM (local time) due to local emissions [55,56]. Pollen that is not being released during one day caused by unfavourable conditions (e.g., during rainy days) is carried over to the following day and released if conditions are favourable, in a similar way as the parameterisation in the atmospheric model COSMO-ART [57]. Loss of catkins sometimes caused by rough weather can reduce the expected pollen emission after the events. This process is neglected in our study. Such simplification is primarily forced by limited knowledge but it is acceptable due to the focus of the study. The emission factor that connects the amount of birch tree cover/m$^2$ with pollen release is based on calibration runs in the same way as Hamaoui-Laguel et al. [58] used the CHIMERE model. This is based on current recommendations on the use of regional scale models [59,60] as current source maps for pollen generally need calibration with pollen data for the study year at hand. Calibration was carried out for the entire pollen season using a setup with only domain D01 and D02 in order to keep computational costs low. After each run, the relative difference was calculated between modelled and observed seasonal pollen integral, as defined by Galan et al. [36], and the emission factor was correspondingly adjusted. After 3 calibration runs the difference between simulated and observed pollen integral was less than 20% for Worcester and the emission factor was accepted for use in the model runs with the three domains D01, D02 and D03.

Anthropogenic emissions are global data sets providing annual emissions of PM$_{2.5}$ and PM$_{10}$ from the EDGAR v 4.3 database [61] where all sources are lumped into one group. Monthly, daily and hourly emissions are based on aggregated MACC emission variations for PM$_{2.5}$, with all sectors lumped together [62]. This is a simplification compared to

air quality studies including both aerosols and chemical transformation. However, it is sufficient as the focus of the study is emission and transport of natural emitted aerosols and potential feedback mechanisms using the extended GOCART scheme. Natural emissions, dynamically driven by meteorology, cover dust, sea-salt and pollen. Pollen emissions in this study are divided into two groups: Birch pollen emission for Europe, which are calculated by variable 1 in the GOCART extension and birch pollen emission for UK, which are calculated by variable 10. The sum of the pollen emission therefore corresponds to total emission, allowing the numerical separation into long distance transport and local (UK) emissions by using the GOCART extension in WRF-Chem both with and without radiative feedback from aerosols.

### 2.5. Evaluation of the Results

Model results are presented in a form of vertical profiles with wind speeds and are compared with observations at the two sounding sites (Figure 2). This approach is often used in modelling studies as an alternative to a more detailed post-processing method involving interpolation of both model results and observations to a fixed set of pre-selected heights [63]. Time series with daily mean concentrations of birch (*Betula*) pollen (with and without feedback) are compared with observations at the 6 monitoring sites (Figure 3) series with daily mean concentrations of birch (*Betula*) pollen (with and without feedback) and compared with observations at the 6 monitoring sites (Figure 3). The pollen time series are separated into emission sources into UK and emissions without UK. Maps (Figure 4) with hourly mean concentrations of birch (*Betula*) pollen are produced with 12 h time intervals during the central part of the episode (29 March–1 April) and the difference between the scenario with and without radiative feedback is calculated for both *Betula* pollen (Figure 5) and total accumulated dust (Figure 6). Total concentration of *Betula* pollen and dust pollen with 6-h intervals are presented for detailed understanding. The difference in daily birch pollen concentrations between the scenario with radiative feedback and without radiative feedback is tested for significance ($p = 0.05$) assuming the difference between the two variables is normally distributed. The results are presented in map-form with two possible outcomes at each grid cell: significant difference or no significance.

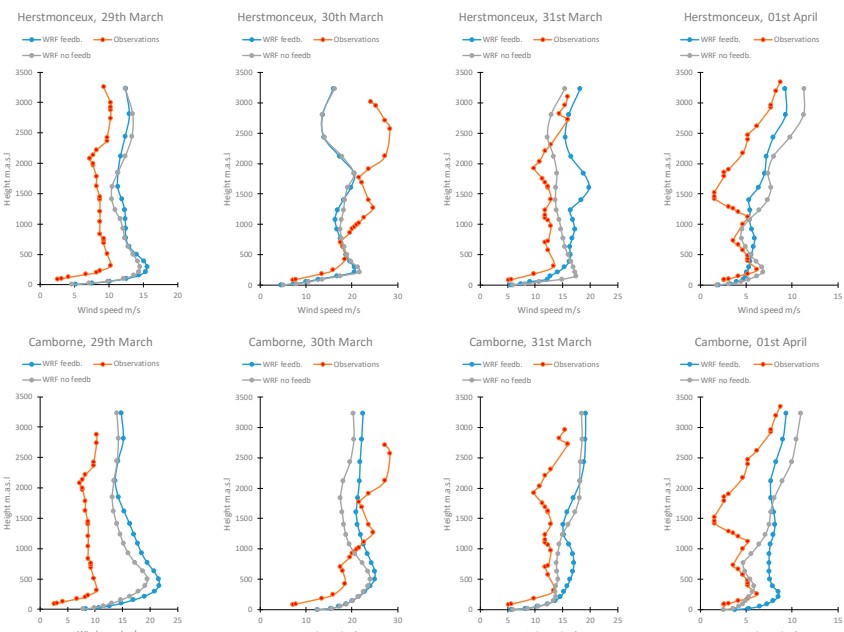

**Figure 2.** Observed vertical profile of wind speed (red line) from the two atmospheric sounding sites and calculated vertical profile of wind speed in the WRF model (domain03) with (blue line) and without (grey line) radiative feedback. The upper panel corresponds to the Herstmonceux site, the lower panel—to the Camborne site.

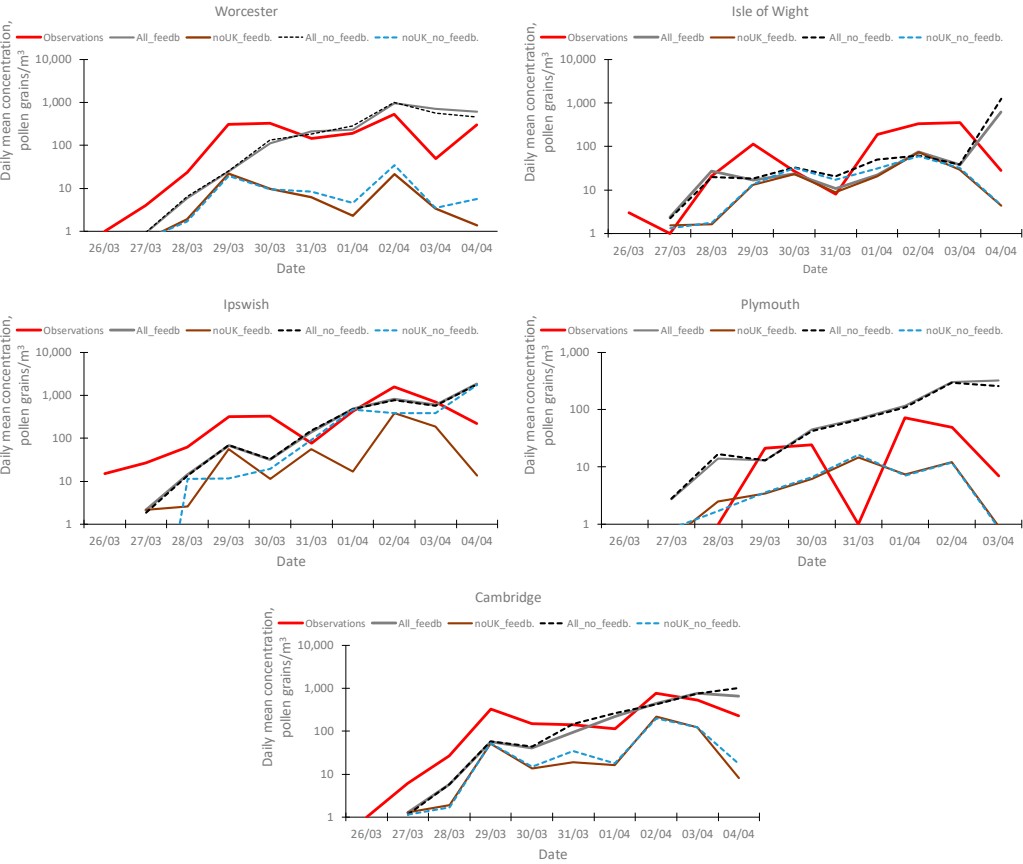

**Figure 3.** Calculated birch pollen concentrations at the 6 sites, with and without radiation feedback, and splitting using two scenarios: without the UK pollen sources and two scenarios with all pollen sources included. The calculated concentrations are plotted versus the observations (red line). Note the y-axis is logarithmic in order to highlight differences at low concentrations.

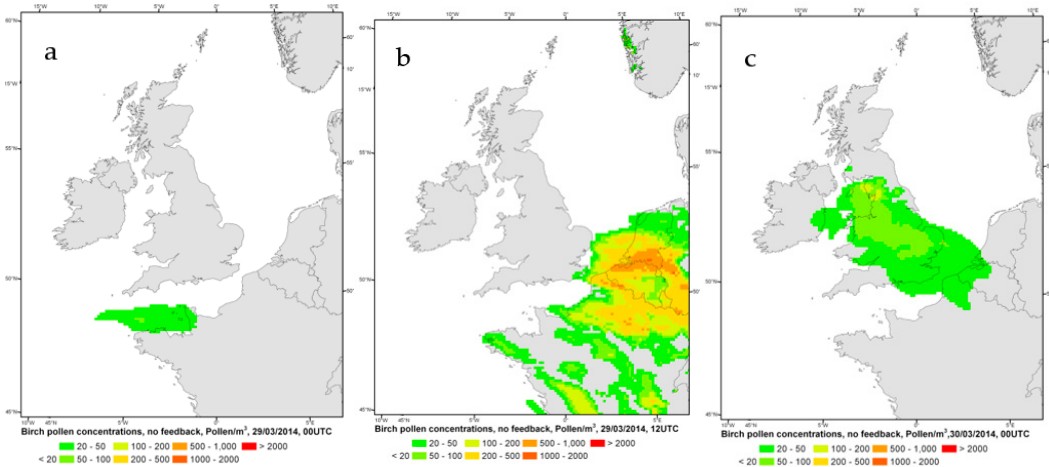

**Figure 4.** *Cont.*

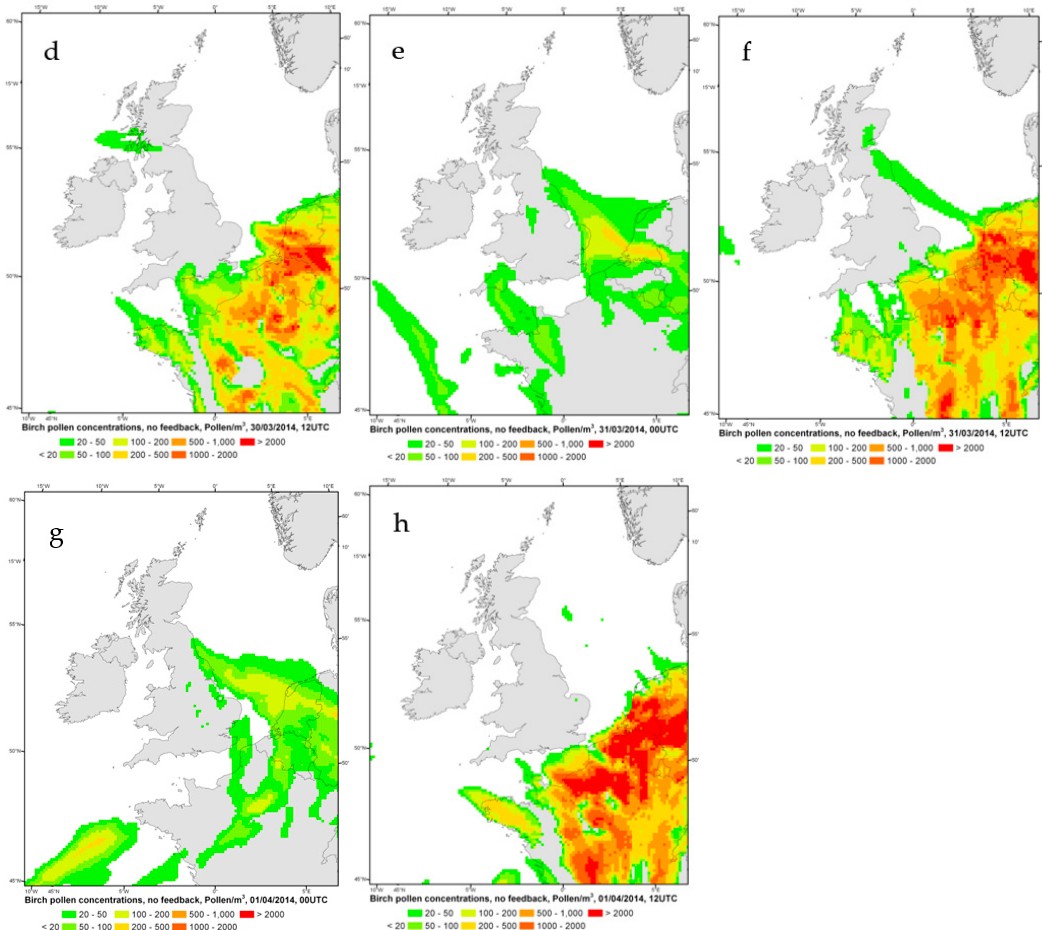

**Figure 4.** Calculated birch pollen concentration (pollen/m³) without UK emission and without radiation feedback during four days (**a**–**h**) 29 March–1 April 2014 with a 12-hour interval going from the top left to lower right.

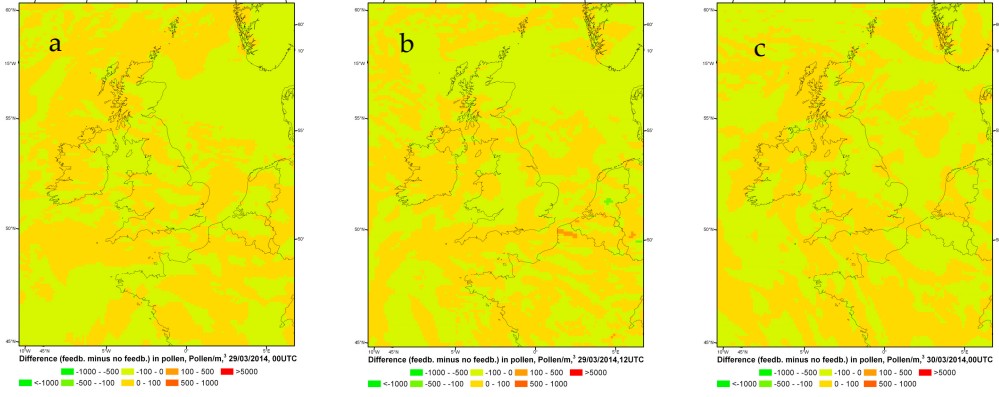

**Figure 5.** *Cont*.

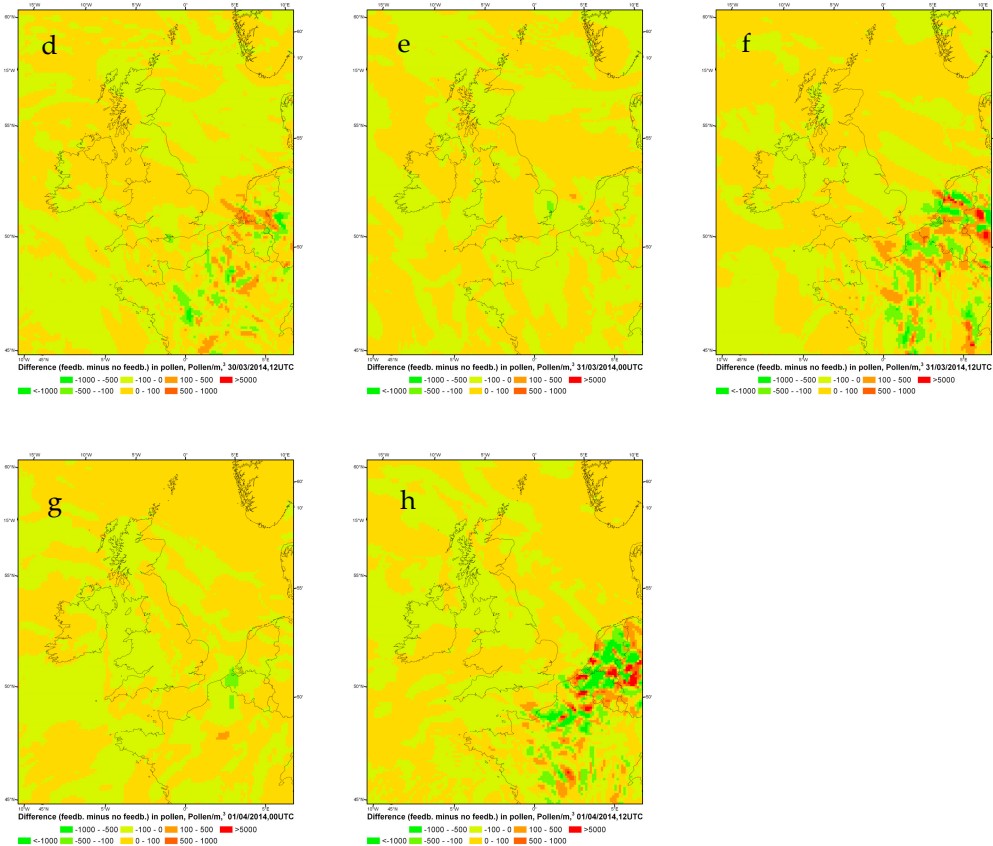

**Figure 5.** Calculated difference in birch pollen concentration (pollen/m³) between the two options with radiation feedback and without radiation feedback during four days (**a**–**h**) 29 March–1 April 2014 with a 12-h interval going from the top left to lower right. The results are based on the WRF-Chem calculations using the emission scenario without UK emissions.

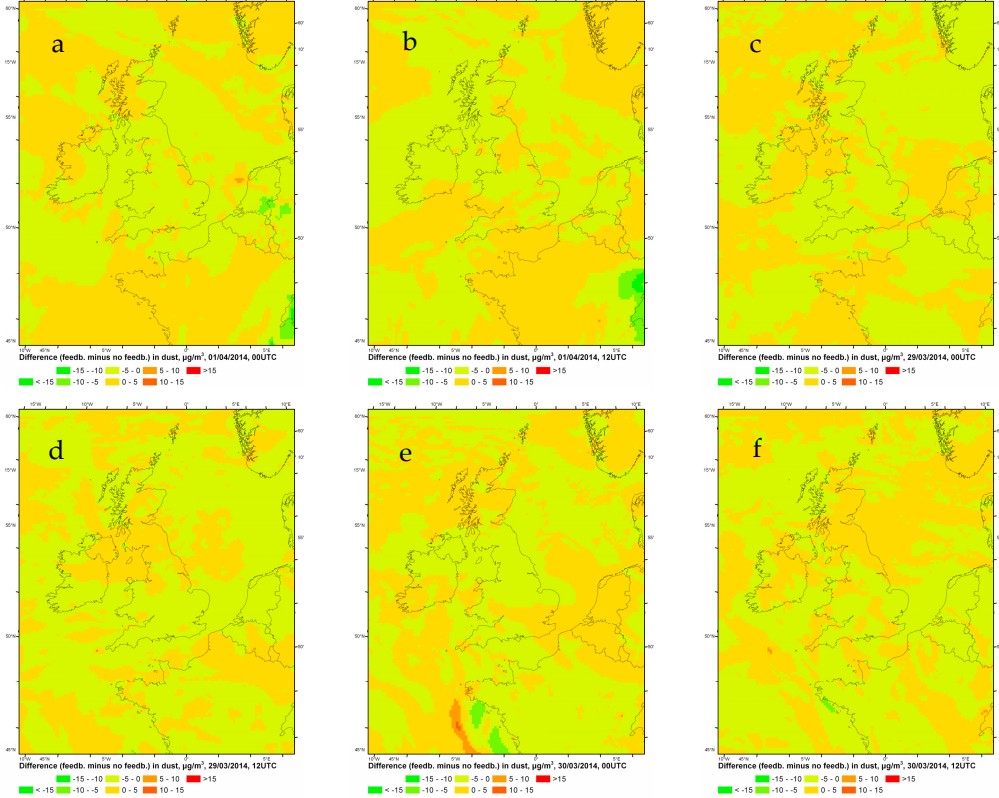

**Figure 6.** *Cont*.

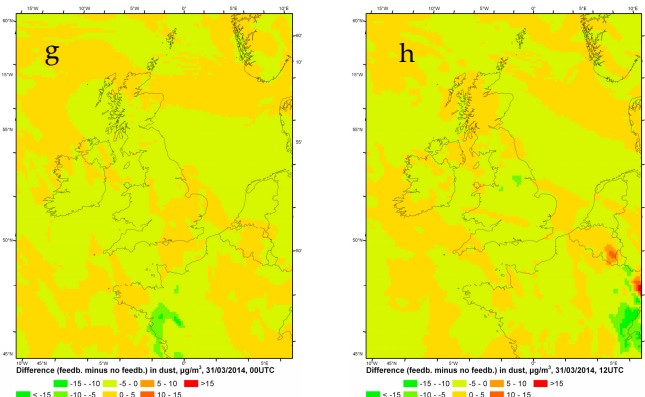

**Figure 6.** Calculated difference in dust concentrations (ug/m$^3$) using the option with radiation feedback and without radiation feedback during four days (**a–h**) 29 March–1 April 2014 with a 12-h interval going from the top left to lower right.

## 3. Results

### 3.1. Pollen Data and Expert Estimates, which Days the Pollen Data Suggest Long Distance Transport

Unusual pollen events, that are indicative of LDT, occurred during the episode as follows (Table 1): an unusually high *Betula* count of 1562 occurred at Ipswich on 1 April 2014. This was the highest peak at this site since it started monitoring in 2011. Meanwhile, other sites had high but relatively lower amounts of *Betula* pollen; *Quercus* pollen was found out-of-season at Worcester, Plymouth and Ipswich throughout much of the period (Table 2). Other tree pollen during that period are *Corylus/Alnus* (Table S1) and *Salix/Fraxinus* (Table S2).

**Table 2.** Observed pollen concentrations in 2014 at the Isle of Wight (IoW), Worcester (Wor), Plymouth (Ply), Cambridge (Cam) and Ipswich (Ips) for *Ulmus* (elm) and *Platanus* (plane).

| Date | *Ulmus* Sp. | | | | | *Platanus* Sp. | | | | |
|---|---|---|---|---|---|---|---|---|---|---|
| **25/3** | **IoW** | **Wor** | **Ply** | **Cam** | **Ips** | **IoW** | **Wor** | **Ply** | **Cam** | **Ips** |
| 26/3 | 0 | 1 | 1 | 0 | 2 | 0 | 0 | 0 | 0 | 0 |
| 27/3 | 0 | 0 | 0 | 0 | 0 | 0 | 0 | 0 | 0 | 2 |
| 28/3 | 0 | 0 | 1 | 0 | 1 | 0 | 0 | 0 | 0 | 11 |
| 29/3 | 1 | 0 | 0 | 0 | 0 | 0 | 0 | 0 | 0 | 2 |
| 30/3 | 3 | 1 | 5 | 0 | 0 | 0 | 1 | 0 | 0 | 1 |
| 31/3 | 0 | 1 | 3 | 0 | 0 | 1 | 7 | 1 | 5 | 9 |
| 1/4 | 1 | 3 | 0 | 0 | 0 | 0 | 1 | 0 | 1 | 12 |
| 2/4 | 1 | 0 | 4 | 0 | 0 | 1 | 0 | 1 | 0 | 6 |
| 3/4 | 0 | 0 | 2 | 0 | 0 | 1 | 5 | 2 | 2 | 8 |
| 4/4 | 2 | 0 | 0 | 0 | 0 | 0 | 1 | 0 | 10 | 8 |
| 5/4 | 0 | 0 | 0 | 0 | 2 | 0 | 0 | 0 | 0 | 0 |

Other pollen types were found by an expert palynologist at the Worcester site that are also indicative of LDT. Firstly, an *Ephedra* sp. pollen grain was observed at 02:00 on 31st March as part of routine counting. This pollen type is associated only with arid, desert habitats and cannot grow in the temperate UK climate. Increased mineral material was observed on the microscope slides at Worcester during the episode, notably from 09:00 on 29th March until 08:00 on 5th April, with a peak period of deposition occurring from 02:00 to 22:00 on 31st March. This period was subjected to additional microscope scanning to find any other pollen types that could have arrived from the Saharan region. This additional analysis revealed the following: Amaranthaceae (x1), *Artemisia* sp. (x1) and *Paronychia* sp. (x2), all of which are found in arid regions. Although types of Amaranthaceae and *Artemisia* can occur in the UK, their season is late summer, not early spring. *Paronychia* sp. is a desert species and cannot grow in the UK.

### 3.2. Vertical Structure of the Atmosphere: Observations vs. Model Calculations with and without Feedback

The vertical profiles in wind speed (Figure 2) for Herstmonceux and Camborne, covering the period 29 March–1 April, show a rapid increase in wind speed within the first few hundred meters from the surface and for most of the period also a smaller decrease again after 500–1000 m. The profile in WRF calculations is similar to the observed, but generally overestimated with a few meters per second, i.e., less than 5 m/s for most profiles. For most of the selected period there is limited impact from feedback in the horizontal wind speeds and in most cases less than a few meters per second. Further exploration of the vertical profiles (see supplementary information with all data) shows for many profiles an increase in temperature and decrease in humidity within the lowest few hundred meters, hence causing an inversion. This suggests that the air mass a few hundred meters up in the atmosphere has a different origin compared to the surface and that this origin is likely from Southern Europe. Additionally, strong dry winds in this air mass combined with an inversion limits downwards turbulence and transport of material through the inversion layer, which causes near ideal conditions for LDT of aerosols found in the dry air mass.

### 3.3. Observed and Simulated Birch Pollen Concentrations, Both Local (UK) and LDT

The model simulations show similar patterns in birch concentrations to the observations (Figure 3), here illustrated with a logarithmic scale to capture the full variability. The results generally show that the pollen grains are observed one day earlier than in the model simulations. The model simulations suggest a very large contribution from non-UK sources during the episode for Plymouth and a smaller contribution to the Isle of Wight, whereas the daily contribution from non-UK sources varies from day to at Worcester, Cambridge and Ipswich (note that these three sites all are located further north than the three other sites). For all the days the difference in birch pollen concentrations between simulation with and without feedback is within ±30%, except for five cases where there is an increase of 52–104%.

### 3.4. Simulated Birch Pollen Concentrations, Both Local (UK) and LDT

The model simulations without UK contributions, i.e., local emissions turned off in the model, (Figure 4) show that for only one of the 8 selected 12 h snapshots (Figure 4c), during the period 29 March–1 April is there a considerable import from the continent, while small areas of the most southern parts of UK have significantly influence on two additional snapshots.

The difference in calculated birch pollen concentration (pollen/m$^3$) with and without radiative feedback (Figure 5) and by excluding the UK sources, show that the impact in most cases is less than 100 pollen/m$^3$ over the UK areas, while the source region (parts of Belgium, Netherlands and France) experience differences up to 1000 grains/m$^3$, noting an almost checkerboard-based positive-negative pattern, potentially originating from a spatial displacement of the pollen grains at the meso-beta scale (20 to 200 km), here using the definition by Orlanski et al. [64].

Dust concentrations (Figure 6) show the impact of up to 5 μg/m$^3$ from feedback mechanisms for the selected snapshots for the entire model domain, noting that most of the dust source in this case is expected to originate from the Sahara.

Total simulated dust concentrations with 6 h intervals during the 31st of March (Figure 7), show a plume reaching most of the UK. However, the daytime (12 and 18 UTC, Figure 7) snapshots show an added and significant contribution of pollen originating from the continent. It should be noted that the pollen contribution is very small during night-time (0 and 06 UTC).

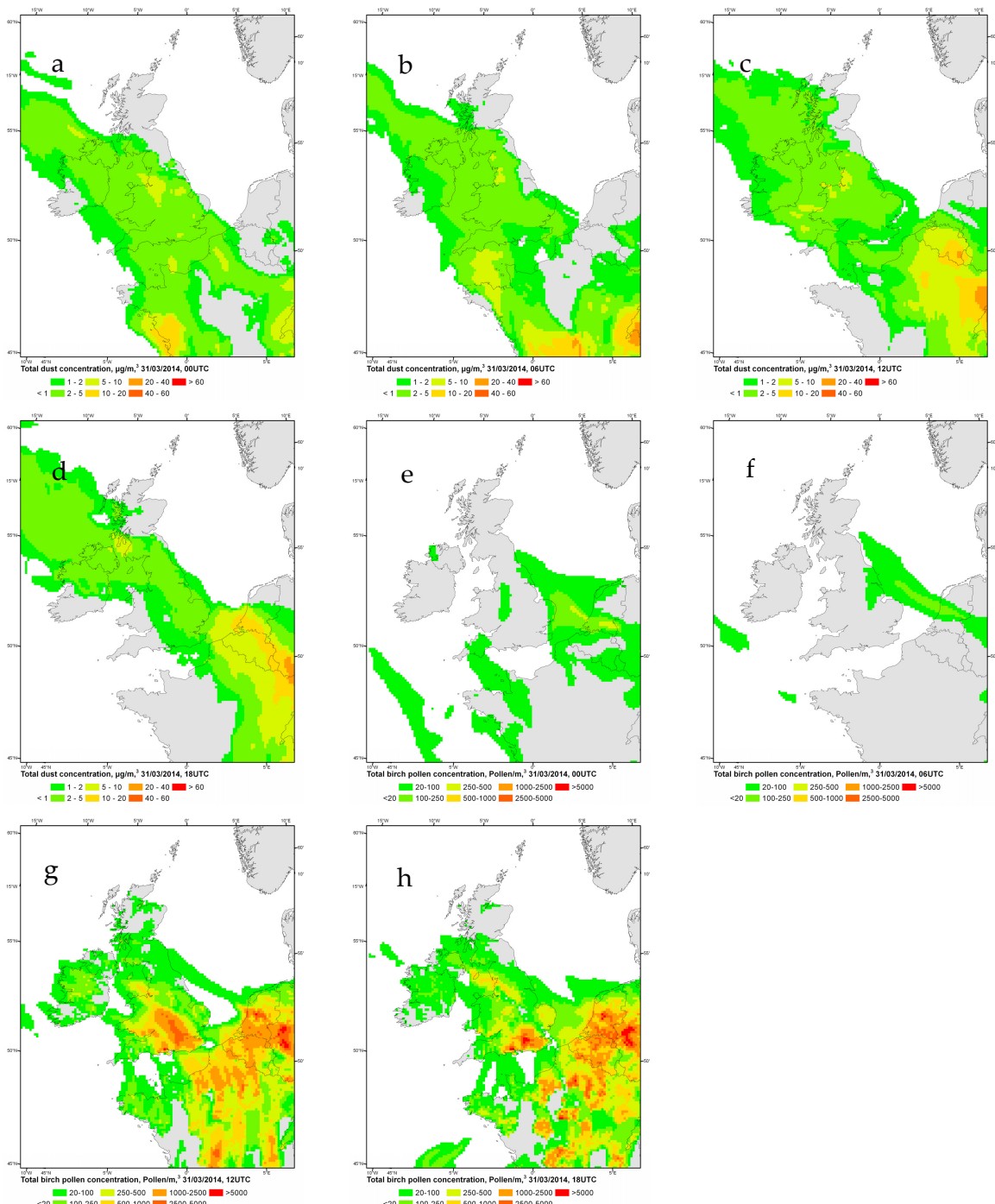

**Figure 7.** Calculated total dust concentrations (ug/m$^3$) (**a**–**d**) and total birch pollen concentration (**e**–**h**) (pollen/m$^3$) (all emissions included) and with radiative feedback during the central period of the episode the 31st of March 2014 with 6 h intervals going from left to right (top: dust, bottom: pollen).

Overall contribution from feedback mechanisms on the daily mean concentrations are statistically significant (*p*-value = 0.05) for nearly the entire model domain (Figure 8), for each of the days, noting that hour to hour concentrations vary from 0 to more than 5000 grains/m$^3$ depending on the location and the time of the day (Figure 7).

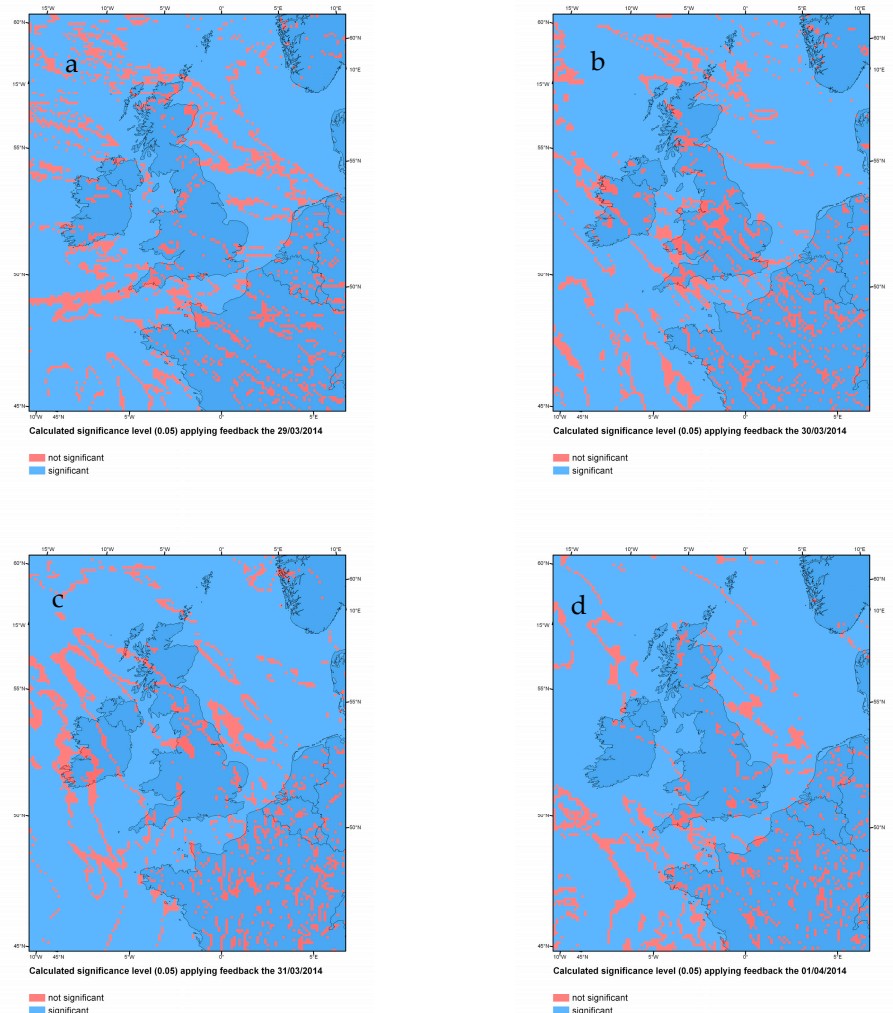

**Figure 8.** Calculated significance (blue) and no significance (red) with respect to the difference in daily mean pollen concentrations in applying radiative feedback vs. non-radiative feedback and for each of the days (**a**–**d**) 29 March–1 April, going from left to right. The results are based on the WRF-Chem calculations using the emission scenario without UK birch pollen emissions. The results are statistically significant with *p*-value = 0.05.

## 4. Discussion and Conclusions

Our study reveals the importance of radiative feedbacks on pollen concentrations by affecting meteorology. In most cases the difference was ± 30% in pollen concentrations, while increased concentrations with more than 100% were simulated. To the best of our knowledge this is the first ever attempt at applying an online atmospheric dispersion model, here WRF-Chem, to estimate the effect of the feedbacks on both local pollen and long distance transport of pollen. In this case we focused on an event previously studied with respect to chemical air pollution and PM [65], where it was found that increased concentrations of PM arrived in two plumes to the UK, partly caused by anthropogenic emissions and partly by Saharan dust. In our study we find that the impact from air pollution on meteorology, through direct feedback processes, has a statistically significant impact on pollen concentration and that the most important spring aeroallergen, birch (*Betula*) pollen, showed unusually high concentrations during the air pollution episode (Figure 3 and Table 1). This was complemented with import of pollen types that were out-of-season for the UK, from Southern Europe and a few pollen types found only in arid regions. This means that those that suffer from allergic rhinitis have been exposed to a harmful cocktail of pollen and aeroallergens (e.g., Figure 7). Elevated air pollution has previously been shown to exacerbate respiratory symptoms [66] among patients and

air pollution has been shown to increase allergenic potency [9]. Now, with our study, air pollution is shown to impact pollen concentrations through feedback mechanisms with meteorology. Saharan dust, relevant for this episode, can have a significant impact on meteorology when the conditions are favourable [67]. Feedback effects on meteorology caused by intense air pollution events have previously been shown to have an important impact on air pollution [32], both due to direct effects on radiation and indirect effects on clouds. These feedback effects have been shown to systematically change forecasting results [32]. Our findings are therefore consistent with these previous results and show that large impacts are found in the areas with large emissions but may also have a statistically significant effect throughout the entire study area.

We found that the model simulates levels of high pollen concentrations consistent with observations during the episode studied, but that the model misses the arrival of the first plume by a one day delay. Errors at this scale or larger are often found using numerical transport models when simulating birch pollen concentrations [27,68]. The reason for the late arrival is probably caused by lack of calibration of the source term in the source area. This is an aspect previously highlighted by several authors [59,60] and for the first part of the simulation most likely related to areas in Northern France, Belgium and The Netherlands (e.g., Figure 7), consistent with findings by Vieno et al [65]. We also find that for most of the episode, the majority of the birch (*Betula*) pollen is due to UK sources (Figures 3 and 4). This suggests that a priority for UK based modelling should be on the UK source term [69], supporting similar conclusions for Belgium [69]. The first plume of Birch pollen is according to the model, entirely caused by LDT, an aspect previously found in both Denmark [70] and Germany [71], here highlighting the usefulness of transport models to predict the start of the season. Predicting the start of the birch pollen season accurately is very valuable to hay fever patients sensitized to birch pollen, allowing them to mitigate the impact of these aeroallergens. Our results suggest that WRF-Chem is a suitable tool for predicting LDT to the UK from the continent.

The meteorological observations as well as the WRF-Chem model show a vertical structure of the atmosphere that involves an inversion and strong dry winds (Figure 2) which favours LDT but also suppresses transport to the surface, driven by turbulence. This was found to be very important for the beginning of the episode keeping the Saharan dust elevated above the surface [65]. In the case of pollen, the UK pollen record shows a range of different pollen types, some of them in smaller quantities. The origin of these pollen could be local but also could have a source in the Netherlands, Belgium, France or maybe even Spain depending on the origin of the air mass. This is one explanation for the presence of pollen in the UK record found outside its normal season. For example oak, which flowers later in April or May would be able to reach UK as the air mass passing Spain and France passed areas where this type was flowering [72]. Here the presence of the LDT component in the pollen record in late March or the beginning of April contradicts the finding by Vieno et al. [65] in which most air pollution was kept elevated and away from the surface. The most likely reason is the relatively large settling velocity of pollen (~1 cm) compared to the much smaller settling velocity for fine particulate matter [73] enabling a fraction of the pollen grains to reach the surface. Here the complex pattern of the air masses with Saharan dust enabled collection of pollen from countries to the south of the UK, typically with an advanced pollen season compared to the UK. It also enabled capture of pollen from areas to the east or northeast with later pollen seasons compared to the southern UK. This suggests that during such episodes, there is a likelihood for arrival of a larger range of pollen to arrive in the UK, augmenting the UK pollen concentrations, where the type of pollen that arrives will depend on the seasonality in the source region.

Overall, during episodes of Saharan dust the atmosphere is in a state that favours long distance transport of PM and aeroallergens from a large geographical region. The high concentration of PM (Figure 5) affects the radiative balance which in turn significantly affects pollen concentrations (Figure 8), increasing co-exposure of pollen and air pollutants. The impact is in most cases statistically significant, but the largest impact, and hence

clinically most important one, is found in the areas with considerable pollen emission causing high concentrations (Figures 5 and 6). A number of studies have repeatedly discussed the impact of emission on local pollen concentrations [59,74]. This underlines that during such severe episodes, the impact of air pollution on meteorology should not be neglected as it can be expected to have a significant contribution to pollen concentrations in areas with pollen emission and to some extent on the long distance components.

**Supplementary Materials:** The following are available online at https://www.mdpi.com/article/10.3390/atmos12111376/s1. Table S1: Observed pollen concentrations in 2014 at Isle of Wight (IoW), Worcester (Wor), Plymouth (Ply), Cambridge (Cam) and Ipswich (Ips) for Corylus (hazel) and Alnus (alder). Unusual pollen counts are highlighted; Table S2: Observed pollen concentrations in 2014 at Isle of Wight (IoW), Worcester (Wor), Plymouth (Ply), Cambridge (Cam) and Ipswich (Ips) for Salix (willow) and Fraxinus (ash).

**Author Contributions:** C.A.S., M.W., M.G. and A.K. contributed to the WRF-Chem model developments, analysis of the results, and production of graphical outputs and paper writing. B.A.-G. contributed to the pollen data collection and extra pollen analysis, analysis of results and paper writing. B.A.-G., C.A.S. and M.W. contributed to the conceptualization about LDT, Saharan dust and feedback in atmospheric modelling based on experiencing the episode in 2014. All authors have read and agreed to the published version of the manuscript.

**Funding:** This research was funded by the European Commission through a Marie Curie Career Integration Grant (Project ID CIG631745, Acronym SUPREME) and a Marie Curie Fellowship (Grant ID: 701753, Acronym IPLATFORM).

**Data Availability Statement:** Raw data of simulated and observed pollen concentrations and the vertical profiles from the different scenarios are available from the public accessible archive WRaP at Worcester University: https://eprints.worc.ac.uk/11424/ (accessed on 6 October 2021). Raw data from atmospheric soundings can be retrieved from the public archive at University of Wyoming: http://weather.uwyo.edu/upperair/sounding.html (accessed on 6 October 2021). Raw data for air pollutant emissions can be obtained from the Edgar data base: http://edgar.jrc.ec.europa.eu/overview.php?v=431 (accessed on 6 October 2021).

**Acknowledgments:** Thanks to the UK Met Office for providing access to pollen data from the UK nationwide monitoring programme. Thanks to "National Centers for Environmental Prediction" for providing access to NCEP FNL Operational Model Global Tropospheric Analyses, https://doi.org/10.5065/D6M043C6.

**Conflicts of Interest:** The authors declare no conflict of interest.

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
