# Peer review of "Air Pollution Affecting Pollen Concentrations through Radiative Feedback in the Atmosphere"

_atmosphere, doi:10.3390/atmos12111376_

Round 1
Reviewer 1 Report
The study is quite interesting, well-designed and in my opinion, fits the journal. This work estimates feedback between meteorology and particles on concentrations of aeroallergens using an extended version of the atmospheric model WRF-Chem and shows that the atmospheric conditions favored meteorological feedback mechanisms altering long distance transport of air pollution and aeroallergens. In general, a very interesting, well designed and very relevant work.
I will present some minor comments that can be considered and further discussed:
Line 53-58. The text says “This suggests that air pollution episodes, in particular those with high particulate matter associated with transport of desert dust, could simultaneously cause LDT of several different aeroaller gens and impact mesoscale meteorology, significantly affecting the concentrations of the 58 aeroallergens involved.” The hypothesis is not clear to me. could you rephrase it? How high particulate matter associated with other pollutants particles could simultaneously cause LDT?
Table 2. Ulmus (elm) and Platanus (plane) are not indicated in the table. Also please indicate the pollen concentration [pollen/m3] here and in table 1.
Figure 2. Very interesting. However, to assess the observed transboundary air pollution path and height related to wind profiles, it would be great if this figure showed also the correlation coefficients between wind speed in each layer and PM10 or PM2.5 for every episode, indicating the various relationships between surface particulate matter and vertical wind profiles.
Low quality of the figures 4-8. The axes or the legends can not be read. Is it possible to improve the resolution?
Line 368. A reference is missing.
Line 384. In my opinion, the importance that this model could have in predicting the start of the pollen season should be emphasized here and in the abstract, since it can significantly affect allergy patients.
Line 398-400. Why here the settling velocity of pollen was large than in Vieno et al? any hypothesis?
Table S2 (supplementary material). Salix (willow) and Fraxinus (ash) are not indicated in the table.
Reviewer 2 Report
I have only two points for improving this paper, the first is to clearly state the significance for conducting this paper in introduction part, at current manuscript, the reasons and necessities for using the data and methods should be further clarified, some recent publication can be referenced, for example, A High-Precision Aerosol Retrieval Algorithm (HiPARA) for Advanced Himawari Imager (AHI) data: Development and verificationï¼› Anthropogenic and meteorological drivers of 1980-2016 trend in aerosol optical and radiative properties over the Yangtze River Basinï¼› Aerosol Optical Properties and Associated Direct Radiative Forcing over the Yangtze River Basin during 2001-2015. The second is to simplify the main conclusions.
Round 2
Reviewer 2 Report
This manuscript has generally improved to some extent, so I think it can be accepted for publication in this journal.